# CENTRAL MOMENT DISCREPANCY (CMD) FOR DOMAIN-INVARIANT REPRESENTATION LEARNING

**Werner Zellinger, Edwin Lughofer & Susanne Saminger-Platz**[*]
Department of Knowledge-Based Mathematical Systems
Johannes Kepler University Linz, Austria
{werner.zellinger, edwin.lughofer, susanne.saminger-platz}@jku.at

**Thomas Grubinger & Thomas Natschläger**[†]
Data Analysis Systems
Software Competence Center Hagenberg, Austria
{thomas.grubinger, thomas.natschlaeger}@scch.at

## ABSTRACT

The learning of domain-invariant representations in the context of *domain adaptation* with neural networks is considered. We propose a new regularization method that minimizes the domain-specific latent feature representations directly in the hidden activation space. Although some standard distribution matching approaches exist that can be interpreted as the matching of weighted sums of moments, e.g. Maximum Mean Discrepancy, an explicit order-wise matching of higher order moments has not been considered before. We propose to match the higher order central moments of probability distributions by means of order-wise moment differences. Our model does not require computationally expensive distance and kernel matrix computations. We utilize the equivalent representation of probability distributions by moment sequences to define a new distance function, called Central Moment Discrepancy (CMD). We prove that CMD is a metric on the set of probability distributions on a compact interval. We further prove that convergence of probability distributions on compact intervals w. r. t. the new metric implies convergence in distribution of the respective random variables. We test our approach on two different benchmark data sets for object recognition (Office) and sentiment analysis of product reviews (Amazon reviews). CMD achieves a new state-of-the-art performance on most domain adaptation tasks of Office and outperforms networks trained with Maximum Mean Discrepancy, Variational Fair Autoencoders and Domain Adversarial Neural Networks on Amazon reviews. In addition, a post-hoc parameter sensitivity analysis shows that the new approach is stable w. r. t. parameter changes in a certain interval. The source code of the experiments is publicly available[1].

## 1 INTRODUCTION

The collection and preprocessing of large amounts of data for new domains is often time consuming and expensive. This in turn limits the application of state-of-the-art methods like deep neural network architectures, that require large amounts of data. However, often data from related domains can be used to improve the prediction model in the new domain. This paper addresses the particularly important and challenging domain-invariant representation learning task of unsupervised domain adaptation (Glorot et al., 2011; Li et al., 2014; Pan et al., 2011; Ganin et al., 2016). In unsupervised domain adaptation, the training data consists of labeled data from the source domain(s) and unlabeled data from the target domain. In practice, this setting is quite common, as in many applications

---

[*]http://www.flll.jku.at
[†]http://www.scch.at
[1]https://github.com/wzell/cmd

the collection of input data is cheap, but the collection of labels is expensive. Typical examples include image analysis tasks and sentiment analysis, where labels have to be collected manually.

Recent research shows that domain adaptation approaches work particularly well with (deep) neural networks, which produce outstanding results on some domain adaptation data sets (Ganin et al., 2016; Sun & Saenko, 2016; Li et al., 2016; Aljundi et al., 2015; Long et al., 2015; Li et al., 2015; Zhuang et al., 2015; Louizos et al., 2016). The most successful methods have in common that they encourage similarity between the latent network representations w. r. t. the different domains. This similarity is often enforced by minimizing a certain distance between the networks' domain-specific hidden activations. Three outstanding approaches for the choice of the distance function are the *Proxy $\mathcal{A}$-distance* (Ben-David et al., 2010), the *Kullback-Leibler* (KL) *divergence* Kullback & Leibler (1951), applied to the mean of the activations (Zhuang et al., 2015), and the *Maximum Mean Discrepancy* (Gretton et al., 2006, MMD).

Two of them, the MMD and the KL-divergence approach, can be viewed as the matching of statistical moments. The KL-divergence approach is based on mean (first raw moment) matching. Using the Taylor expansion of the Gaussian kernel, most MMD-based approaches can be viewed as minimizing a certain distance between weighted sums of all raw moments (Li et al., 2015).

The interpretation of the KL-divergence approaches and MMD-based approaches as moment matching procedures motivate us to match the higher order moments of the domain-specific activation distributions directly in the hidden activation space. The matching of the higher order moments is performed explicitly for each moment order and each hidden coordinate. Compared to KL-divergence-based approaches, which only match the first moment, our approach also matches higher order moments. In comparison to MMD-based approaches, our method explicitly matches the moments for each order, and it does not require any computationally expensive distance- and kernel matrix computations.

The proposed distribution matching method induces a metric between probability distributions. This is possible since distributions on compact intervals have an equivalent representation by means of their moment sequences. We utilize central moments due to their translation invariance and natural geometric interpretation. We call the new metric Central Moment Discrepancy (CMD).

The contributions of this paper are as follows:

- We propose to match the domain-specific hidden representations by explicitly minimizing differences of higher order central moments for each moment order. We utilize the equivalent representation of probability distributions by moment sequences to define a new distance function, which we call Central Moment Discrepancy (CMD).

- Probability theoretic analysis is used to prove that CMD is a metric on the set of probability distributions on a compact interval.

- We additionally prove that convergence of probability distributions on compact intervals w. r. t. to the new metric implies convergence in distribution of the respective random variables. This means that minimizing the CMD metric between probability distributions leads to convergence of the cumulative distribution functions of the random variables.

- In contrast to MMD-based approaches our method does not require computationally expensive kernel matrix computations.

- We achieve a new state-of-the-art performance on most domain adaptation tasks of Office and outperform networks trained with MMD, variational fair autoencoders and domain adversarial neural networks on Amazon reviews.

- A parameter sensitivity analysis shows that CMD is insensitive to parameter changes within a certain interval. Consequently, no additional hyper-parameter search has to be performed.

## 2 HIDDEN ACTIVATION MATCHING

We consider the *unsupervised domain adaptation* setting (Glorot et al., 2011; Li et al., 2014; Pan et al., 2011; Ganin et al., 2016) with an input space $\mathcal{X}$ and a label space $\mathcal{Y}$. Two distributions over $\mathcal{X} \times \mathcal{Y}$ are given: the *labeled source domain* $D_S$ and the *unlabeled target domain* $D_T$. Two corresponding samples are given: the *source sample* $S = (X_S, Y_S) = \{(x_i, y_i)\}_{i=1}^n \overset{\text{i.i.d.}}{\sim} (D_S)^n$ and

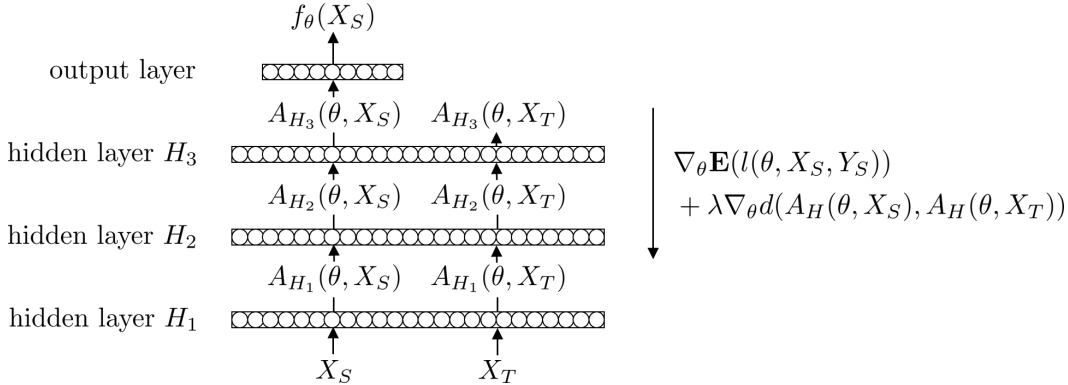

Figure 1: Schematic sketch of a three layer neural network trained with backpropagation based on objective (2). $\nabla_\theta$ refers to the gradient w. r. t. $\theta$.

the *target sample* $T = X_T = \{x_i\}_{i=1}^m \overset{\text{i.i.d.}}{\sim} (D_T)^m$. The goal of the unsupervised domain adaptation setting is to build a classifier $f : \mathcal{X} \to \mathcal{Y}$ with a low target risk $R_T(f) = \Pr_{(x,y)\sim D_T} (f(x) \neq y)$, while no information about the labels in $D_T$ is given.

We focus our studies on neural network classifiers $f_\theta : \mathcal{X} \to \mathcal{Y}$ with parameters $\theta \in \Theta$, the input space $\mathcal{X} = \mathbb{R}^I$ with input dimension $I$, and the label space $\mathcal{Y} = [0,1]^{|C|}$ with the cardinality $|C|$ of the set of classes $C$. We further assume a network output $f_\theta(x) \in [0,1]^{|C|}$ of an example $x \in \mathbb{R}^I$ to be normalized by the softmax-function $\sigma : \mathbb{R}^{|C|} \to [0,1]^{|C|}$ with $\sigma(z)_j = \frac{e^{z_j}}{\sum_{k=1}^{|C|} e^{z_k}}$ for $z = \{z_1, \ldots, z_{|C|}\}$. We focus on bounded activation functions $g_H : \mathbb{R} \to [a,b]^N$ for the hidden layer $H$ with $N$ hidden nodes, e.g. the hyperbolic tangent or the sigmoid function. Unbounded activation functions, e.g. rectified linear units or exponential linear units, can be used if the output is clipped or normalized to be bounded. Using the loss function $l : \Theta \times \mathcal{X} \times \mathcal{Y} \to \mathbb{R}$, e.g. cross-entropy $l(\theta, x, y) = -\sum_{i \in C} y_i \log(f_\theta(x)_i)$, and the sample set $(X, Y) \subset \mathbb{R}^I \times [0,1]^{|C|}$, we define the objective function as

$$\min_{\theta \in \Theta} \mathbf{E}(l(\theta, X, Y)) \tag{1}$$

where $\mathbf{E}$ denotes the empirical expectation, i.e. $\mathbf{E}(l(\theta, X, Y)) = \frac{1}{|(X,Y)|} \sum_{(x,y) \in (X,Y)} l(\theta, x, y)$. Let us denote the *source hidden activations* by $A_H(\theta, X_S) = g_H(\theta_H^T A_{H'}(\theta, X_S)) \subset [a,b]^N$ and the *target hidden activations* by $A_H(\theta, X_T) = g_H(\theta_H^T A_{H'}(\theta, X_T)) \subset [a,b]^N$ for the hidden layer $H$ with $N$ hidden nodes and parameter $\theta_H$, and the hidden layer $H'$ before $H$.

One fundamental assumption of most unsupervised domain adaptation networks is that the source risk $R_S(f)$ is a good indicator for the target risk $R_T(f)$, when the domain-specific latent space representations are similar (Ganin et al., 2016). This similarity can be enforced by matching the distributions of the hidden activations $A_H(\theta, X_S)$ and $A_H(\theta, X_T)$ of higher layers $H$. Recent state-of-the-art approaches define a domain regularizer $d : ([a,b]^N)^n \times ([a,b]^N)^m \to [0,\infty)$, which gives a measure for the domain discrepancy in the activation space $[a,b]^N$. The domain regularizer is added to the objective by means of an additional weighting parameter $\lambda$.

$$\min_{\theta \in \Theta} \mathbf{E}(l(\theta, X_S, Y_S)) + \lambda \cdot d(A_H(\theta, X_S), A_H(\theta, X_T)) \tag{2}$$

Fig. 1 shows a sketch of the described architecture and fig. 2 shows the hidden activations of a simple neural network optimized by eq. (1) (left) and eq. (2) (right). It can be seen that similar activation distributions are obtained when being optimized on the basis of the domain regularized objective.

## 3 RELATED WORK

Recently, several measures $d$ for objective (2) have been proposed. One approach is the *Proxy $\mathcal{A}$-distance*, given by $\hat{d}_\mathcal{A} = 2(1 - 2\epsilon)$, where $\epsilon$ is the generalization error on the problem of discriminating between source and target samples (Ben-David et al., 2010). Ganin et al. (2016) compute the

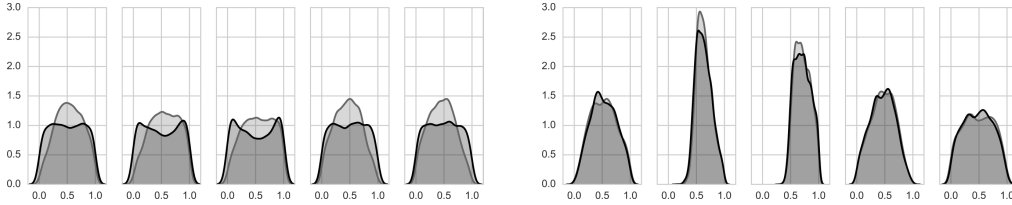

Figure 2: Hidden activation distributions for a simple one-layer classification network with sigmoid activation functions and five hidden nodes trained with the standard objective (1) (left) and objective (2) that includes the domain discrepancy minimization (right). The approach of this paper was used as domain regularizer. Dark gray: activations of the source domain, light gray: activations of the target domain.

value $\epsilon$ with a neural network classifier that is simultaneously trained with the original network by means of a gradient reversal layer. They call their approach *domain-adversarial neural networks*. Unfortunately, a new classifier has to be trained in this approach including the need of new parameters, additional computation times and validation procedures.

Another approach is to make use of the MMD (Gretton et al., 2006) as domain regularizer.

$$\text{MMD}(X,Y)^2 = \mathbf{E}(K(X,X)) - 2\mathbf{E}(K(X,Y)) + \mathbf{E}(K(Y,Y)) \tag{3}$$

where $\mathbf{E}(K(X,Y)) = \frac{1}{|X| \cdot |Y|} \sum_{k \in K(X,Y)} k$ is the empirical expectation of the kernel products $k$ between all examples in $X$ and $Y$ stored by the kernel matrix $K(X,Y)$. A suitable choice of the kernel seems to be the Gaussian kernel $e^{-\beta \|x-y\|^2}$ (Louizos et al., 2016; Li et al., 2015; Tzeng et al., 2014). This approach has two major drawbacks: (a) the need of tuning an additional kernel parameter $\beta$, and (b) the need of the kernel matrix computation $K(X,Y)$ (computational complexity $\mathcal{O}(n^2 + nm+m^2)$), which becomes inefficient (resource-intensive) in case of large data sets. Concerning (a), the tuning of $\beta$ is sophisticated since no target samples are available in the domain adaptation setting. Suitable tuning procedures are transfer learning specific cross-validation methods (Zhong et al., 2010). More general methods that don't utilize source labels include heuristics that are based on kernel space properties (Sriperumbudur et al., 2009; Gretton et al., 2012), combinations of multiple kernels (Li et al., 2015), and kernel choices that maximize the MMD test power (Sutherland et al., 2016). The drawback (b) of the kernel matrix computation can be handled by approximating the MMD (Zhao & Meng, 2015), or by using linear time estimators (Gretton et al., 2012). In this work we focus on the quadratic-time MMD with the Gaussian kernel (Gretton et al., 2012; Tzeng et al., 2014) and transfer learning specific cross-validation for parameter tuning (Zhong et al., 2010; Ganin et al., 2016).

The two approaches MMD and the *Proxy $\mathcal{A}$-distance* have in common that they do not minimize the domain discrepancy explicitly in the hidden activation space. In contrast, the authors in Zhuang et al. (2015) do so by minimizing a modified version of the *Kullback-Leibler divergence* of the mean activations (MKL). That is, for samples $X, Y \subset \mathbb{R}^N$,

$$\text{MKL}(X,Y) = \sum_{i=1}^{N} \mathbf{E}(X)_i \log \frac{\mathbf{E}(X)_i}{\mathbf{E}(Y)_i} + \mathbf{E}(Y)_i \log \frac{\mathbf{E}(Y)_i}{\mathbf{E}(X)_i} \tag{4}$$

with $\mathbf{E}(X)_i$ being the $i^{\text{th}}$ coordinate of the empirical expectation $\mathbf{E}(X) = \frac{1}{|X|} \sum_{x \in X} x$. This approach is fast to compute and has an explicit interpretation in the activation space. Our empirical observations (section *Experiments*) show that minimizing the distance between only the first moment (mean) of the activation distributions can be improved by also minimizing the distance between higher order moments.

As noted in the introduction, our approach is motivated by the fact that the MMD and the KL-divergence approach can be seen as the matching of statistical moments of the hidden activations $A_H(\theta, X_S)$ and $A_H(\theta, X_T)$. In particular, MMD-based approaches that use the Gaussian kernel are equivalent to minimizing a certain distance between weighted sums of all moments of the hidden activation distributions (Li et al., 2015).

We propose to minimize differences of higher order central moments of the activations $A_H(\theta, X_S)$ and $A_H(\theta, X_T)$. The difference minimization is performed explicitly for each moment order. Our

approach utilizes the equivalent representation of probability distributions in terms of its moment series. We further utilize central moments due to their translation invariance and natural geometric interpretation. Our approach contrasts with other moment-based approaches, as they either match only the first moment (MKL) or they don't explicitly match the moments for each order (MMD). As a result, our approach improves over MMD-based approaches in terms of computational complexity with $\mathcal{O}\left(N(n+m)\right)$ for CMD and $\mathcal{O}\left(N(n^2+nm+m^2)\right)$ for MMD. In contrast to MKL-based approaches more accurate distribution matching characteristics are obtained. In addition, CMD achieves a new state-of-the-art performance on most domain adaptation tasks of Office and outperforms networks trained with MMD, variational fair autoencoders and domain adversarial neural networks on Amazon reviews.

## 4    CENTRAL MOMENT DISCREPANCY (CMD)

In this section we first propose a new distance function CMD on probability distributions on compact intervals. The definition is extended by two theorems that identify CMD as a metric and analyze a convergence property. The final domain regularizer is then defined as an empirical estimate of CMD. The proofs of the theorems are given in the appendix.

**Definition 1** (CMD metric). *Let $X = (X_1, \ldots, X_n)$ and $Y = (Y_1, \ldots, Y_n)$ be bounded random vectors independent and identically distributed from two probability distributions $p$ and $q$ on the compact interval $[a,b]^N$. The central moment discrepancy metric (CMD) is defined by*

$$CMD(p,q) = \frac{1}{|b-a|} \left\| \mathbb{E}(X) - \mathbb{E}(Y) \right\|_2 + \sum_{k=2}^{\infty} \frac{1}{|b-a|^k} \left\| c_k(X) - c_k(Y) \right\|_2 \qquad (5)$$

*where $\mathbb{E}(X)$ is the expectation of $X$, and*

$$c_k(X) = \left( \mathbb{E}\Big( \prod_{i=1}^{N} (X_i - \mathbb{E}(X_i))^{r_i} \Big) \right)_{\substack{r_1 + \ldots + r_N = k \\ r_1, \ldots, r_n \geq 0}}$$

*is the central moment vector of order $k$.*

The first order central moments are zero, the second order central moments are related to variance, and the third and fourth order central moments are related to the skewness and the kurtosis of probability distributions. It is easy to see that $\mathrm{CMD}(p,q) \geq 0$, $\mathrm{CMD}(p,q) = \mathrm{CMD}(q,p)$, $\mathrm{CMD}(p,q) \leq \mathrm{CMD}(p,r) + \mathrm{CMD}(r,q)$ and $p = q \Rightarrow \mathrm{CMD}(p,q) = 0$. The following theorem shows the remaining property for CMD to be a metric on the set of probability distributions on a compact interval.

**Theorem 1.** *Let $p$ and $q$ be two probability distributions on a compact interval and let CMD be defined as in (5), then*

$$CMD(p,q) = 0 \;\; \Rightarrow \;\; p = q$$

Our approach is to minimize the discrepancy between the domain-specific hidden activation distributions by minimizing the CMD. Thus, in the optimization procedure, we increasingly expect to see the domain-specific cumulative distribution functions approach each other. This characteristic can be expressed by the concept of *convergence in distribution* and it is shown in the following theorem.

**Theorem 2.** *Let $p_n$ and $p$ be probability distributions on a compact interval and let CMD be defined as in (5), then*

$$CMD(p_n, p) \to 0 \;\; \Rightarrow \;\; p_n \overset{d}{\to} p$$

*where $\overset{d}{\to}$ denotes convergence in distribution.*

We define the final *central moment discrepancy regularizer* as an empirical estimate of the CMD metric. Only the central moments that correspond to the marginal distributions are computed. The number of central moments is limited by a new parameter $K$ and the expectation is sampled by the empirical expectation.

**Definition 2** (CMD regularizer). *Let $X$ and $Y$ be bounded random samples with respective probability distributions $p$ and $q$ on the interval $[a, b]^N$. The central moment discrepancy regularizer $CMD_K$ is defined as an empirical estimate of the CMD metric, by*

$$CMD_K(X, Y) = \frac{1}{|b-a|} \|\mathbf{E}(X) - \mathbf{E}(Y)\|_2 + \sum_{k=2}^{K} \frac{1}{|b-a|^k} \|C_k(X) - C_k(Y)\|_2 \qquad (6)$$

*where $\mathbf{E}(X) = \frac{1}{|X|} \sum_{x \in X} x$ is the empirical expectation vector computed on the sample $X$ and $C_k(X) = \mathbf{E}((x - \mathbf{E}(X))^k)$ is the vector of all $k^{th}$ order sample central moments of the coordinates of $X$.*

This definition includes three approximation steps: (a) the computation of only marginal central moments, (b) the bound on the order of central moment terms via parameter $K$, and (c) the sampling of the probability distributions by the replacement of the expected value with the empirical expectation.

Applying approximation (a) and assuming independent marginal distributions, a zero CMD distance value still implies equal joint distributions (thm. 1) but convergence in distribution (thm. 2) applies only to the marginals. In the case of dependent marginal distributions, zero CMD distance implies equal marginals and convergence in CMD implies convergence in distribution of the marginals. However, the matching properties for the joint distributions are not obtained with dependent marginals and approximation (a). The computational complexity is reduced to be linear w. r. t. the number of samples.

Concerning (b), proposition 1 shows that the marginal distribution specific CMD terms have an upper bound that is strictly decreasing with increasing moment order. This bound is convergent to zero. That is, higher CMD terms can contribute less to the overall distance value. This observation is experimentally strengthened in subsection *Parameter Sensitivity*.

**Proposition 1.** *Let $X$ and $Y$ be bounded random vectors with respective probability distributions $p$ and $q$ on the compact interval $[a, b]^N$. Then*

$$\frac{1}{|b-a|^k} \|c_k(X) - c_k(Y)\|_2 \leq 2\sqrt{N} \left( \frac{1}{k+1} \left( \frac{k}{k+1} \right)^k + \frac{1}{2^{1+k}} \right) \qquad (7)$$

*where $c_k(X) = \mathbb{E}((X - \mathbb{E}(X))^k)$ is the vector of all $k^{th}$ order sample central moments of the marginal distributions of $p$.*

Concerning approximation (c), the joint application of the weak law of large numbers (Billingsley, 2008) with the continuous mapping theorem (Billingsley, 2013) proves that this approximation creates a consistent estimate.

We would like to underline that the training of neural networks with eq. (2) and the CMD regularizer in eq. (6) can be easily realized by gradient descent algorithms. The gradients of the CMD regularizer are simple aggregations of derivatives of the standard functions $g_H$, $x^k$ and $\|.\|_2$.

## 5 EXPERIMENTS

Our experimental evaluations are based on two benchmark datasets for domain adaptation, *Amazon reviews* and *Office*, described in subsection *Datasets*. The experimental setup is discussed in subsection *Experimental Setup* and our classification accuracy results are discussed in subsection *Results*. Subsection *Parameter Sensitivity* analysis the accuracy sensitivity w. r. t. parameter changes of $K$ for CMD and $\beta$ for MMD.

### 5.1 DATASETS

**Amazon reviews:** For our first experiment we use the *Amazon reviews* data set with the same preprocessing as used by Chen et al. (2012); Ganin et al. (2016); Louizos et al. (2016). The data set contains product reviews of four different product categories: books, DVDs, kitchen appliances and electronics. Reviews are encoded in 5000 dimensional feature vectors of bag-of-words unigrams and bigrams with binary labels: 0 if the product is ranked by $1 - 3$ stars and 1 if the product is ranked

by $4$ or $5$ stars. From the four categories we obtain twelve domain adaptation tasks (each category serves once as source category and once as target category).

**Office:** The second experiment is based on the computer vision classification data set from Saenko et al. (2010) with images from three distinct domains: *amazon* (A), *webcam* (W) and *dslr* (D). This data set is a *de facto* standard for domain adaptation algorithms in computer vision. Amazon, the largest domain, is a composition of 2817 images and its corresponding 31 classes. Following previous works we assess the performance of our method across all six possible transfer tasks.

## 5.2 EXPERIMENTAL SETUP

**Amazon Reviews:**

For the Amazon reviews experiment, we use the same data splits as previous works for every task. Thus we have 2000 labeled source examples and 2000 unlabeled target examples for training, and between 3000 and 6000 examples for testing.

We use a similar architecture as Ganin et al. (2016) with one dense hidden layer with 50 hidden nodes, sigmoid activation functions and softmax output function. Three neural networks are trained by means of eq. (2): (a) a base model without domain regularization ($\lambda = 0$), (b) with the MMD as domain regularizer and (c) with CMD as domain regularizer. These models are additionally compared with the state-of-the-art models VFAE (Louizos et al., 2016) and DANN (Ganin et al., 2016). The models (a),(b) and (c) are trained with similar setup as in Louizos et al. (2016) and Ganin et al. (2016).

For the CMD regularizer, the $\lambda$ parameter of eq. (2) is set to 1, i.e. the weighting parameter $\lambda$ is neglected. The parameter $K$ is heuristically set to five, as the first five moments capture rich geometric information about the shape of a distribution and $K = 5$ is small enough to be computationally efficient. However, the experiments in subsection *Parameter Sensitivity* show that similar results are obtained for $K \geq 3$.

For the MMD regularizer we use the Gaussian kernel with parameter $\beta$. We performed a hyperparameter search for $\beta$ and $\lambda$, which has to be performed in an unsupervised way (no labels in the target domain). We use a variant of the *reverse cross-validation* approach proposed by Zhong et al. (2010), in which we initialize the model weights of the reverse classifier by the weights of the first learned classifier (see Ganin et al. (2016) for details). Thereby, the parameter $\lambda$ is tuned on 10 values between 0.1 and 500 on a logarithmic scale. The parameter $\beta$ is tuned on 10 values between 0.01 and 10 on a logarithmic scale. Without this parameter search, no competitive prediction accuracy results could be obtained.

Since we have to deal with sparse data, we rely on the *Adagrad* optimizer (Duchi et al., 2011). For all evaluations, the default parametrization is used as implemented in *Keras* (Chollet, 2015). All evaluations are repeated 10 times based on different shuffles of the data, and the mean accuracies and standard deviations are analyzed.

**Office:** Since the office dataset is rather small with only 2817 images in its largest domain, we use the latent representations of the convolution neural network VGG16 of Simonyan & Zisserman (2014). In particular we train a classifier with one hidden layer, 256 hidden nodes and sigmoid activation function on top of the output of the first dense layer in the network. We again train one base model without domain regularization and a CMD regularized version with $K = 5$ and $\lambda = 1$.

We follow the standard training protocol for this data set and use all available source and target examples during training. Using this "fully-transductive" protocol, we compare our method with other state-of-the-art approaches including DLID (Chopra et al., 2013), DDC (Tzeng et al., 2014), DAN (Long et al., 2015), Deep CORAL (Sun & Saenko, 2016), and DANN (Ganin et al., 2016), based on fine-tuning of the baseline model AlexNet (Krizhevsky et al., 2012). We further compare our method to LSSA (Aljundi et al., 2015), CORAL (Sun et al., 2016), and AdaBN (Li et al., 2016), based on the fine-tuning of InceptionBN (Ioffe & Szegedy, 2015).

As an alternative to Adagrad for non-sparse data, we use the *Adadelta* optimizer from Zeiler (2012). Again, the default parametrization from Keras is used. We handle unbalances between source and target sample by randomly down-sampling (up-sampling) the source sample. In addition, we ensure a sub-sampled source batch that is balanced w. r. t. the class labels.

Since all hyper-parameters are set a-priori, no hyper-parameter search has to be performed.

All experiments are repeated 10 times with randomly shuffled data sets and random initializations.

## 5.3 RESULTS

**Amazon Reviews:** Table 1 shows the classification accuracies of four models: The *Source Only* model is the non domain regularized neural network trained with objective (1), and serves as a base model for the domain adaptation improvements. The models MMD and CMD are trained with the same architecture and objective (2) with $d$ as the domain regularizer MMD and CMD, respectively. VFAE refers to the Variational Fair Autoencoder of Louizos et al. (2016), including a slightly modified version of the MMD regularizer for faster computations, and DANN refers to the domain-adversarial neural networks model of Ganin et al. (2016). The last two columns are taken directly from these publications.

As one can observe in table 1, our accuracy of the CMD-based model is the highest in 9 out of 12 domain adaptation tasks, whereas on the remaining 3 it is the second best method. However, the difference in accuracy compared to the best method is smaller than the standard deviation over all data shuffles.

Table 1: Prediction accuracy $\pm$ standard deviation on the Amazon reviews dataset. The last two columns are taken directly from Louizos et al. (2016) and Ganin et al. (2016).

| Source→Target | Source Only | MMD | CMD | VFAE | DANN |
|---|---|---|---|---|---|
| books→dvd | $.787 \pm .004$ | $.796 \pm .008$ | $\mathbf{.805 \pm .007}$ | .799 | .784 |
| books→electronics | $.714 \pm .009$ | $.758 \pm .018$ | $.787 \pm .007$ | **.792** | .733 |
| books→kitchen | $.745 \pm .006$ | $.787 \pm .019$ | $.813 \pm .008$ | **.816** | .779 |
| dvd→books | $.746 \pm .019$ | $.780 \pm .018$ | $\mathbf{.795 \pm .005}$ | .755 | .723 |
| dvd→electronics | $.724 \pm .011$ | $.766 \pm .025$ | $\mathbf{.797 \pm .010}$ | .786 | .754 |
| dvd→kitchen | $.765 \pm .012$ | $.796 \pm .019$ | $\mathbf{.830 \pm .012}$ | .822 | .783 |
| electronics→books | $.711 \pm .006$ | $.733 \pm .017$ | $\mathbf{.744 \pm .008}$ | .727 | .713 |
| electronics→dvd | $.719 \pm .009$ | $.748 \pm .013$ | $.763 \pm .006$ | **.765** | .738 |
| electronics→kitchen | $.844 \pm .005$ | $.857 \pm .007$ | $\mathbf{.860 \pm .004}$ | .850 | .854 |
| kitchen→books | $.699 \pm .014$ | $.740 \pm .017$ | $\mathbf{.756 \pm .006}$ | .720 | .709 |
| kitchen→dvd | $.734 \pm .011$ | $.763 \pm .011$ | $\mathbf{.775 \pm .005}$ | .733 | .740 |
| kitchen→electronics | $.833 \pm .004$ | $.844 \pm .007$ | $\mathbf{.854 \pm .003}$ | .838 | .843 |
| average | $.752 \pm .009$ | $.781 \pm .015$ | $\mathbf{.798 \pm .007}$ | .784 | .763 |

**Office:** Table 2 shows the classification accuracy of different models trained on the Office dataset. Note that some of the methods (LSSA, CORAL and AdaBN) are evaluated based on the Inception BN model, which shows higher accuracy than the base model (VGG16) of our method in most tasks. However, our method outperforms related state-of-the-art methods on all except two tasks, on which it performs similar. We improve the previous state-of-the-art method AdaBN (Li et al., 2016) by more than 3.2% in average accuracy.

## 5.4 PARAMETER SENSITIVITY

The first sensitivity experiment aims at providing evidence regarding the accuracy sensitivity of the CMD regularizer w. r. t. parameter changes of $K$. That is, the contribution of higher terms in the CMD regularizer are analyzed. The claim is that the accuracy of CMD-based networks does not depend strongly on the choice of $K$ in a range around its default value 5.

In fig. 3 on the upper left we analyze the classification accuracy of a CMD-based network trained on all tasks of the Amazon reviews experiment. We perform a grid search for the two regularization hyper-parameters $\lambda$ and $K$. We empirically choose a representative stable region for each parameter, $[0.3, 3]$ for $\lambda$ and $\{1, \ldots, 7\}$ for $K$. Since we want to analyze the sensitivity w. r. t. $K$, we averaged over the $\lambda$-dimension, resulting in one accuracy value per $K$ for each of the 12 tasks. Each accuracy is transformed into an accuracy ratio value by dividing it with the accuracy of $K = 5$. Thus, for each $K$ and task we get one value representing the ratio between the obtained accuracy (for this $K$ and task) and the accuracy of $K = 5$. The results are shown in fig. 3 (upper left). The accuracy

Table 2: Prediction accuracy $\pm$ standard deviation on the Office dataset. The first 10 rows are taken directly from the papers of Ganin et al. (2016) and Li et al. (2016). The models DLID –DANN are based on the AlexNet model, LSSA –AdaBN are based on the InceptionBN model, and our method (CMD) is based on the VGG16 model.

| Method | A→W | D→W | W→D | A→D | D→A | W→A | average |
|---|---|---|---|---|---|---|---|
| AlexNet | .616 | .954 | .990 | .638 | .511 | .498 | .701 |
| DLID | .519 | .782 | .899 | - | - | - | - |
| DDC | .618 | .950 | .985 | .644 | .521 | .522 | .707 |
| Deep CORAL | .664 | .957 | .992 | .668 | .528 | .515 | .721 |
| DAN | .685 | .960 | .990 | .670 | .540 | .531 | .729 |
| DANN | .730 | **.964** | .992 | - | - | - | - |
| InceptionBN | .703 | .943 | **1.00** | .705 | .601 | .579 | .755 |
| LSSA | .677 | .961 | .984 | .713 | .578 | .578 | .749 |
| CORAL | .709 | .957 | .998 | .719 | .590 | .602 | .763 |
| AdaBN | .742 | .957 | .998 | .731 | .598 | .574 | .767 |
| VGG16 | $.676 \pm .006$ | $.961 \pm .003$ | $.992 \pm .002$ | $.739 \pm .009$ | $.582 \pm .005$ | $.578 \pm .004$ | .755 |
| CMD | $\mathbf{.770 \pm .006}$ | $.963 \pm .004$ | $.992 \pm .002$ | $\mathbf{.796 \pm .006}$ | $\mathbf{.638 \pm .007}$ | $\mathbf{.633 \pm .006}$ | **.799** |

ratios between $K = 5$ and $K \in \{3, 4, 6, 7\}$ are lower than $0.5\%$, which underpins the claim that the accuracy of CMD-based networks does not depend strongly on the choice of $K$ in a range around its default value 5. For $K = 1$ and $K = 2$ higher ratio values are obtained. In addition, for these two values many tasks show worse accuracy than obtained by $K \in \{3, 4, 5, 6, 7\}$. From this we additionally conclude that higher values of $K$ are preferable to $K = 1$ and $K = 2$.

The same experimental procedure is performed with MMD regularization wighted by $\lambda \in [5, 45]$ and Gaussian kernel parameter $\beta \in [0.3, 1.7]$. We calculate the ratio values w. r. t. the accuracy of $\beta = 1.2$, since this value of $\beta$ shows the highest mean accuracy of all tasks. Fig. 3 (upper right) shows the results. It can be seen that the accuracy of the MMD network is more sensitive to parameter changes than the CMD regularized version. Note that the problem of finding the best settings for the parameter $\beta$ of the Gaussian kernel is a well known problem (Hsu et al., 2003).

The default number of hidden nodes in all our experiments is 256 because of the high classification accuracy of the networks without domain regularization (Source Only) on the source domains. The question arises if the accuracy of the CMD is lower for higher numbers of hidden nodes. That is, if the accuracy ratio between the accuracy, of the CMD regularized networks compared to the accuracy of the Source Only models, decreases with increasing hidden activation dimension. In order to answer this question we calculate these ratio values for each task of the Amazon reviews data set for different number of hidden nodes $(128, 256, 384, \ldots, 1664)$. For higher numbers of hidden nodes our Source Only models don't converge with the optimization settings under consideration. For the parameters $\lambda$ and $K$ we use our default setting $\lambda = 1$ and $K = 5$. Fig. 3 on the lower left shows the ratio values (vertical axis) for every number of hidden nodes (horizontal axis) and every task (colored lines). It can be seen that the accuracy improvement of the CMD domain regularizer varies between $4\%$ and $6\%$. However, no accuracy ratio decrease can be observed.

Please note that we use a default setting for $K$ and $\lambda$. Thus, fig. 3 shows that our default setting $(\lambda = 1, K = 5)$ can be used independently of the number of hidden nodes. This is an additional result.

The same procedure is performed with the MMD weighted by parameter $\lambda = 9$ and $\beta = 1.2$ as these values show the highest classification accuracy for 256 hidden nodes. Fig. 3 on the lower right shows that the accuracy improvement using the MMD decreases with increasing number of hidden nodes for this parameter setting. That is, for accurate performance of the MMD, additional parameter tuning procedures for $\lambda$ and $\beta$ need to be performed. Note that the problem of finding the best setting for the parameter $\beta$ of the Gaussian kernel is a well known problem (Hsu et al., 2003).

## 6 CONCLUSION AND OUTLOOK

In this paper we proposed the central moment discrepancy (CMD) for domain-invariant representation learning, a distance function between probability distributions. Similar to other state-of-the-art approaches (MMD, KL-*divergence*, *Proxy $\mathcal{A}$-distance*), the CMD function can be used to minimize the domain discrepancy of latent feature representations. This is achieved by order-wise differences

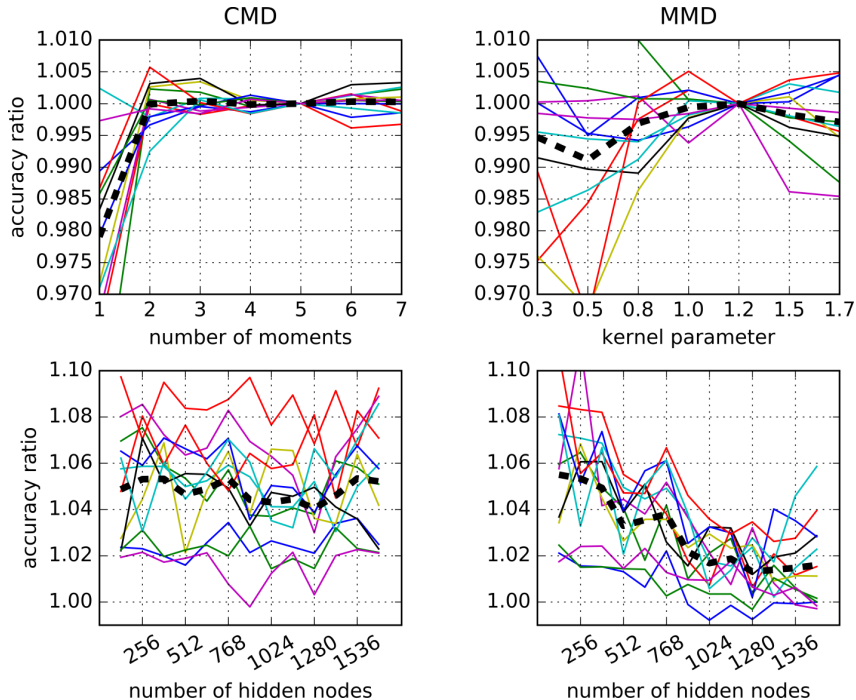

Figure 3: Sensitivity of classification accuracy w. r. t. different parameters of CMD (left) and MMD (right) on the Amazon reviews dataset. The horizontal axes show parameter values and the vertical axes show accuracy ratio values. Each line represents accuracy ratio values for one specific task. The ratio values are computed w. r. t. the default accuracy for CMD (upper left), w. r. t. the best obtainable accuracy for MMD (upper right) and w. r. t. the non domain regularized network accuracies (lower left and lower right).

of central moments. By using probability theoretic analysis, we proved that CMD is a metric and that convergence in CMD implies convergence in distribution for probability distributions on compact intervals. Our method yields state-of-the-art performance on most tasks of the Office benchmark data set and outperforms Gaussian kernel based MMD, VFAE and DANN on most tasks of the Amazon reviews benchmark data set. These results are achieved with the default parameter setting of $K = 5$. In addition, we experimentally underpinned the claim that the classification accuracy is not sensitive to the particular choice of $K$ for $K \geq 3$. Therefore, no computationally expensive hyper-parameter selection is required.

In our experimental analysis we compared our approach to different other state-of-the-art distribution matching methods like the Maximum Mean Discrepancy (MMD) based on the Gaussian kernel using a quadratic time estimate. In the future we want to extend our experimental analysis to other MMD approaches including other kernels, parameter selection procedures and linear time estimators. In addition, we plan to use the CMD for training generative models and to further investigate the approximation quality of the proposed empirical estimate.

## A    THEOREM PROOFS

**Theorem 1.** *Let $p$ and $q$ be two probability distributions on a compact interval and let CMD be defined as in (5), then*

$$CMD(p, q) = 0 \;\; \Rightarrow \;\; p = q$$

*Proof.* Let $X$ and $Y$ be two random vectors that have probability distributions $p$ and $q$, respectively. Let $\hat{X} = X - \mathbb{E}(X)$ and $\hat{Y} = Y - \mathbb{E}(Y)$ be the mean centered random variables. From $CMD(p, q) = 0$ it follows that all moments of the bounded random variables $\hat{X}$ and $\hat{Y}$ are equal. Therefore, the joint moment generating functions of $\hat{X}$ and $\hat{Y}$ are equal. Using the property that $p$

and $q$ have compact support, we obtain the equality of the joint distribution functions of $\hat{X}$ and $\hat{Y}$. Since $\mathbb{E}(X) = \mathbb{E}(Y)$, it follows that $X = Y$. $\qquad\square$

**Theorem 2.** *Let $p_n$ and $p$ be probability distributions on a compact interval and let CMD be defined as in (5), then*

$$CMD(p_n, p) \to 0 \quad \Rightarrow \quad p_n \xrightarrow{d} p$$

*where $\xrightarrow{d}$ denotes convergence in distribution.*

*Proof.* Let $X_n$ and $X$ be random vectors that have probability distributions $p_n$ and $p$ respectively. Let $\hat{X} = X - \mathbb{E}(X)$ and $\hat{X}_n = X_n - \mathbb{E}(X_n)$ be the mean centered random variables. From $CMD(X_n, X) \to 0$ it follows that the moments of $\hat{X}_n$ converge to the moments of $\hat{X}$. Therefore, the joint moment generating functions of $\hat{X}_n$ converge to the joint moment generating function of $\hat{X}$, which implies convergence in distribution of the mean centered random variables. Using $\mathbb{E}(X_n) \to \mathbb{E}(X)$ we obtain $p_n \xrightarrow{d} p$. $\qquad\square$

**Proposition 1.** *Let $X$ and $Y$ be bounded random vectors with respective probability distributions $p$ and $q$ on the compact interval $[a, b]^N$. Then*

$$\frac{1}{|b-a|^k} \|c_k(X) - c_k(Y)\|_2 \le 2\sqrt{N} \left( \frac{1}{k+1} \left( \frac{k}{k+1} \right)^k + \frac{1}{2^{1+k}} \right) \tag{8}$$

*where $c_k(X) = \mathbb{E}((X - \mathbb{E}(X))^k)$ is the vector of all $k^{th}$ order sample central moments of the marginal distributions of $p$.*

*Proof.* Let $\mathcal{X}([a,b])$ be the set of all random variables with values in $[a, b]$. Then it follows that

$$\frac{1}{|b-a|^k} \|c_k(X) - c_k(Y)\|_2 = \left\| \frac{c_k(X)}{|b-a|^k} - \frac{c_k(Y)}{|b-a|^k} \right\|_2$$

$$\le \left\| \frac{c_k(X)}{|b-a|^k} \right\|_2 + \left\| \frac{c_k(Y)}{|b-a|^k} \right\|_2$$

$$= \left\| \mathbb{E}\left( \left( \frac{X - \mathbb{E}(X)}{|b-a|} \right)^k \right) \right\|_2 + \left\| \mathbb{E}\left( \left( \frac{Y - \mathbb{E}(Y)}{|b-a|} \right)^k \right) \right\|_2$$

$$\le \left\| \mathbb{E}\left( \left| \frac{X - \mathbb{E}(X)}{b-a} \right|^k \right) \right\|_2 + \left\| \mathbb{E}\left( \left| \frac{Y - \mathbb{E}(Y)}{b-a} \right|^k \right) \right\|_2$$

$$\le 2\sqrt{N} \sup_{X \in \mathcal{X}([a,b])} \mathbb{E}\left( \left| \frac{X - \mathbb{E}(X)}{b-a} \right|^k \right)$$

The latter term refers to the absolute central moment of order $k$, for which the smallest upper bound is known (Egozcue et al., 2012):

$$\frac{1}{|b-a|^k} \|c_k(X) - c_k(Y)\|_2 \le 2\sqrt{N} \sup_{x \in [0,1]} x(1-x)^k + (1-x)x^k$$

Egozcue et al. (2012) also give a more explicit bound:

$$\frac{1}{|b-a|^k} \|c_k(X) - c_k(Y)\|_2 \le 2\sqrt{N} \left( \frac{1}{k+1} \left( \frac{k}{k+1} \right)^k + \frac{1}{2^{1+k}} \right)$$

$\qquad\square$

ACKNOWLEDGEMENTS

The research reported in this paper has been supported by the Austrian Ministry for Transport, Innovation and Technology, the Federal Ministry of Science, Research and Economy, and the Province of Upper Austria in the frame of the COMET center SCCH.

We would like to thank Bernhard Moser and Florian Sobieczky for fruitful discussions on metric spaces.

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
