# Peer review of "Central Moment Discrepancy (CMD) for Domain-Invariant Representation Learning"

_ICLR 2017 — accepted_

[Official Review · AnonReviewer3 · rating 7 · confidence 4 · 17 Dec 2016]
**CMD for distribution matching.**

This paper proposed a new metric central moment discrepancy (CMD) for matching two distributions, with applications to domain adaptation.  Compared to a more well-known variant, MMD, CMD has the benefit of not over penalizing the mean, and therefore can focus more on the shape of distribution around the center.

In terms of discriminative power (the ability to tell two distributions apart), MMD and CMD should be equivalent, but in practice I can understand that CMD may be better as MMD tries to match the raw moments which may over penalize data that are not zero centered.

In the paper CMD is used only up to Kth order, and not all the central moments are used, but rather only the diagonal entries are considered in the CMD objective, I think this is mostly motivated for computation efficiency.  A natural comparison with MMD therefore can be made, by also explicitly include raw moments up to Kth order.  Another thing to compare against is to include all moments, not just the diagonal terms, in the objective.  This is computationally expensive, but can be done for e.g. 1st and 2nd orders.

Since the experiments only compare CMD in the above form with kernelized MMD, the claim that explicit moment matching is helpful is not very well supported.  To make this a solid claim CMD should be compared against MMD with explicit raw moments.

The claim that the kernel parameter in MMD is hard to tune and CMD does not have such parameters only applies to kernel MMD, not explicit MMD.  For kernel MMD, there are also studies on how to set these parameters, for example:

Sriperumbudur et al.  Kernel choice and classifiability for rkhs embeddings of probability distributions.
Gretton et al.  A kernel two-sample test.

and also using multiple kernels (Li et al. 2015) which removes the need to tune them.  Tuning the beta directly like done in this paper is usually not the way MMD is tuned.  At least simple heuristics like dividing |x-y|^2 by dimensionality or mean pairwise distance first should be applied first before trying beta in the way done in this paper.

Overall I think CMD could be better than MMD, and could have applications in many domains.  But it also has the problem of not easily kernelizable (you can argue this both ways though).  The experiments demonstrating that CMD is better could be done more convincinly.

[Official Review · AnonReviewer2 · rating 9 · confidence 5 · 17 Dec 2016]
**A very simple method with impressive results**

Variational auto-encoders, adversarial networks, and kernel scoring rules like MMD have recently gained popularity as methods for learning directed generative models and for other applications like domain adaptation. This paper gives an additional method along the scoring rules direction that uses the matching of central moments to match two probability distributions. The technique is simple, and in the case of domain adaptation, highly effective.

CMD seems like a very nice and straightforward solution to the domain adaptation problem. The method is computationally straightforward to implement, and seems quite stable with respect to the tuning parameters when compared to MMD. I was skeptical reading through this, especially given the fact that you only use K=5 in your experiments, but the results seem quite good. The natural question that I have now is: how will this method do in training generative models? This is beyond the scope of this paper, but it’s the lowest hanging fruit.

Below I give more detailed feedback.

One way to speed up MMD is to use a random Fourier basis as was done in “Fastmmd: Ensemble of circular discrepancy for efficient two-sample test” by Zhao and Meng, 2015. There are also linear time estimators, e.g., in “A Kernel Two-Sample Test“ by Gretton et al., 2012. I don’t think you need to compare against these approaches since you compare to the full MMD, but they should be cited.

The paper “Generative Models and Model Criticism via Optimized Maximum Mean Discrepancy” by Sutherland et al. submitted to ICLR 2017 as well, discusses techniques for optimizing the kernel used in MMD and is worth citing in section 3.

How limiting is the assumption that the distribution has independent marginals?

The sample complexity of MMD depends heavily on the dimensionality of the input space - do you have any intuitions about the sample complexity of CMD? It seems like it's relatively insensitive based on the results in Figure 4, but I would be surprised if this were the case with 10,000 hidden units. I mainly ask this because with generative models, the output space can be quite high-dimensional.

I’m concerned that the central moments won’t be numerically stable at higher orders when backpropagating. This doesn’t seem to be a problem in the experimental results, but perhaps the authors could comment a bit on this? I’m referring to the fact that ck(X) can be very large for k >= 3. Proposition 1 alleviates my concerns that the overall objective is unstable, I’m referring specifically to the individual terms within.

Figure 3 is rather cluttered, and aside from the mouse class it’s not clear to me from the visualization that the CMD regularizer is actually helping. It would be useful to remove some of the classes for the purpose of visualization.

I would like some clarification about the natural geometric interpretations of K=5. Do you mean that the moments up to K=5 have been well-studied? Do you have any references for this? Why does K >= 6 not have a natural geometric interpretation?

Figure 4 should have a legend

[Official Review · AnonReviewer1 · rating 6 · confidence 4 · 19 Dec 2016]
**Good work with some limitations**

The work introduces a new regularization for learning domain-invariant representations with neural networks. The regularization aims at matching the higher order central moments of the hidden activations of the NNs of the source and target domain. The authors compared the proposed method vs MMD and two state-of-art NN domain adaptation algorithms on the Amazon review and office datasets, and showed comparable performance. 

The idea proposed is simple and straightforward, and the empirical results suggest that it is quite effective. The biggest limitation I can see with the proposed method is the assumption that the hidden activations are independently distributed. For example, this assumption will clearly be violated for the hidden activations of convolutional layers, where neighboring activations are dependent. I guess this is why the authors start with the output of dense layers for the image dataset. Do the authors have insight on if it is beneficial to start adaptation from lower level? If so, do the authors have insight on how to relax the assumption? In these scenarios, if MMD has an advantage as it does not make this assumption? 

Figure 3 does not seems to clearly support the boost of performance shown in table 2. The only class where the new regularization brings the source and target domain closer seem to be the mouse class pointed by the authors. Is the performance improvement only coming from this single class?

[Final Decision · Program Chairs · 06 Feb 2017]
**ICLR committee final decision**

The proposed Central Moment Discrepancy criterion is well-described and supported in this paper. Its performance on domain adaptation tasks is good as well. The reviewers had several good comments and suggestions and the authors have taken most of these into account and improved the paper considerably. The paper thus makes a nice contribution to the distribution matching literature and toolbox.